# Effects of Deoxynivalenol and Fumonisins Fed in Combination to Beef Cattle: Immunotoxicity and Gene Expression

**DOI:** 10.3390/toxins13100714

**Published:** 2021-10-10

**Authors:** Heaven L. Roberts, Massimo Bionaz, Duo Jiang, Barbara Doupovec, Johannes Faas, Charles T. Estill, Dian Schatzmayr, Jennifer M. Duringer

**Affiliations:** 1Department of Animal & Rangeland Sciences, College of Agricultural Sciences, Oregon State University, Corvallis, OR 97331, USA; heavenlearoberts@gmail.com (H.L.R.); Massimo.Bionaz@oregonstate.edu (M.B.); 2Department of Statistics, College of Science, Oregon State University, Corvallis, OR 97331, USA; Duo.Jiang@oregonstate.edu; 3BIOMIN Research Center, Technopark 1, 3430 Tulln, Austria; barbara.doupovec@dsm.com (B.D.); johannes.faas@dsm.com (J.F.); dian.schatzmayr@dsm.com (D.S.); 4Department of Clinical Sciences, Carlson College of Veterinary Medicine, Oregon State University, Corvallis, OR 97331, USA; Charles.Estill@oregonstate.edu; 5Department of Environmental & Molecular Toxicology, College of Agricultural Sciences, Oregon State University, Corvallis, OR 97331, USA

**Keywords:** *Fusarium*, beef cattle, immune function, RNA-Seq, deoxynivalenol, fumonisin

## Abstract

We evaluated the effects of a treatment diet contaminated with 1.7 mg deoxynivalenol and 3.5 mg fumonisins (B1, B2 and B3) per kg ration on immune status and peripheral blood gene expression profiles in finishing-stage Angus steers. The mycotoxin treatment diet was fed for a period of 21 days followed by a two-week washout period during which time all animals consumed the control diet. Whole-blood leukocyte differentials were performed weekly throughout the experimental and washout period. Comparative profiles of CD4^+^ and CD8^+^ T cells, along with bactericidal capacity of circulating neutrophils and monocytes were evaluated at 0, 7, 14, 21 and 35 days. Peripheral blood gene expression was measured at 0, 7, 21 and 35 days via RNA sequencing. Significant increases in the percentage of CD4^−^CD8^+^ T cells were observed in treatment-fed steers after two weeks of treatment and were associated with decreased CD4:CD8 T-cell ratios at this same timepoint (*p* ≤ 0.10). No significant differences were observed as an effect of treatment in terms of bactericidal capacity at any timepoint. Dietary treatments induced major changes in transcripts associated with endocrine, metabolic and infectious diseases; protein digestion and absorption; and environmental information processing (inhibition of signaling and processing), as evaluated by dynamic impact analysis. DAVID analysis also suggested treatment effects on oxygen transport, extra-cellular signaling, cell membrane structure and immune system function. These results indicate that finishing-stage beef cattle are susceptible to the immunotoxic and transcript-inhibitory effects of deoxynivalenol and fumonisins at levels which may be realistically encountered in feedlot situations.

## 1. Introduction

Several metabolites of *Fusarium* fungi, collectively known as *Fusarium* toxins, have demonstrated deleterious effects in humans and livestock that target immune and hepatic systems [1,2]. The *Fusarium* toxins deoxynivalenol (DON) and fumonisins (FUM) often co-occur in animal feed products [3,4,5]. As such, research into simultaneous exposures to these compounds has increased, with special attention given to their long-term effects in livestock species that ultimately enter the human food chain. Ruminants, including cattle, are generally considered tolerant of these compounds, yet numerous studies document toxicity in this group of mammals [6,7,8,9]. For DON, this can include decreased feed consumption and alterations in the rumen such as decreased NDF digestibility; for FUM, changes in liver enzymes and markers related to metabolism have been noted [10]. A review on mycotoxin effects in ruminants cited a need for research in this group of livestock, particularly studies that consider the simultaneous occurrence of dietary mycotoxins, as they can have additive, antagonistic or synergistic effects [10]. 

Behind the toxic potential of these compounds are highly conserved molecular mechanisms. As a trichothecene, DON disrupts transcription and cellular repair through processes which have been previously well reviewed [11]. One of these sequelae—inhibition of protein synthesis—appears to target the immune and gastro-hepatic systems [8,12,13,14,15]. The pathogenesis of dysfunction from fumonisin exposure is linked to the inhibition of serine palmitoyltransferase, which alters sphingolipid metabolism and disrupts cell membranes and function [1,16,17,18]. In addition to the more direct impact of *Fusarium* toxins on target organs, the concurrent alterations of the transcriptome observed in organs [19] and cell cultures [20,21], or a delayed release of these toxins following tissue deposition [22,23], may contribute to long-term damage that remains unresolved even after removal of the contaminated feed source.

This work was part of a larger experiment [24], the purpose of which was to both quantify the effects of dietary DON and FUM co-exposure typically seen in regimens for finishing cattle and evaluate the adequacy of a clearance period in ameliorating these effects. To represent a dose relevant to these economically important production animals, the toxin levels chosen were based on previous surveys of commonly used feedstuffs while not exceeding guidance levels [3,4,5]. In the present study, we utilized a global transcriptomic approach of whole blood to investigate changes in gene expression pathways which may influence long-term animal health through a three-week mycotoxin exposure period, followed by a two-week clearance period. While we examined whole system changes through the Dynamic Impact Approach (DIA) and the Database for Annotation, Visualization and Integrated Discovery (DAVID), we were particularly focused on immune and metabolic pathway alterations. Systemic modulation of immune function was further characterized by flow cytometric and biochemical analysis of peripheral blood throughout the treatment and clearance periods. These data will serve the beef cattle industry’s efforts to understand, quantify, and ultimately overcome the risks associated with *Fusarium* toxins in finishing cattle through the development of targeted management practices.

## 2. Results and Discussion

### 2.1. White Blood Cell Population Dynamics and Functionality

A notable increase in total white blood cells (WBC) was observed in treatment-fed steers beginning at week 3, which continued throughout the remainder of the trial (Figure 1); however, these changes were not significant (*p* > 0.10). Changes in major leukocyte types (lymphocytes, neutrophils, and monocytes), as a percentage of total leukocyte count, were not significantly affected by diet (*p* > 0.10; Figure 1), though numerical patterns reversed for all three major subsets between two and three weeks of treatment. Per the diagnostic laboratory from where these results were obtained, average values for control-fed steers were within a normal range (white blood cell counts 4000–12,000 cells/µL, with lymphocytes 45–75%, monocytes 2–7% and neutrophils 15–45% of white blood cells) at all time points, while total white blood cell counts for treatment-fed animals were, on average, elevated past week one of the trial. In comparative in vitro studies of peripheral blood mononuclear cells obtained from cattle, swine and chickens, Novak et al. (2018) concluded that bovine cells were more sensitive to DON than those from other livestock species [13]. Our results indicate that 1.7 mg DON and 3.5 mg FUM per kg of total ration were insufficient to cause significant cytotoxic effects on circulating mononuclear leukocytes in these animals. 

Neutrophil and monocyte bactericidal functions did not significantly differ between treatment- and control-fed steers throughout the experiment (*p* > 0.10; Figure 2). Interestingly, phagocytic capacity of granulocytes from all steers was substantially reduced by week 5, while monocyte activity began decreasing even earlier in the study (Figure 2).

From the present work, it appears that this may be a natural progression of the fattening of beef steer, which may increase susceptibility to pathogenic bacteria at this critical stage of production. The data presented by Allen et al. (2001) also showed substantially reduced phagocytic activity in finishing-stage steers, regardless of prior mycotoxin exposure. While the effect of body condition or growth status on these outcomes was not formally evaluated in this study, these findings are deserving of future research into the vulnerability of these high-cost, high-demand animals at this state of extreme physiological demand. 

In control-fed steers only, CD8^+^ lymphocytes decreased during the first two weeks of the treatment period (Figure 3), while treatment-fed animals had significantly higher.

CD4^−^CD8^+^ lymphocyte populations during week 2 (*p* ≤ 0.10; Figure 3). This change in the CD8^+^ lymphocyte population ultimately led to significantly reduced CD4:CD8 ratios in treatment-fed steers during the same time period (*p* ≤ 0.10; Figure 3). The decreased CD8^+^ population in control-fed animals is assumed to be an adaptive response to the high-starch diet and increased body condition of control-fed animals of the current study. *Fusarium* toxins such as DON are generally considered pro-inflammatory at low doses or early on in exposure, with anti-inflammatory consequences following prolonged toxicity [11]. Total leukocyte and neutrophil percentages indicate a more grossly sustained pro-inflammatory response, with CD4^−^CD8^+^ lymphocytes and the CD4:CD8 ratio responding more sensitively. Decreased proliferation of CD4^−^CD8^+^, CD4^+^CD8- and CD4^+^CD8^+^ porcine T cells with DON exposure at levels greater than 0.4 µM and reduced proliferation of peripheral blood mononuclear cells (PBMCs, primarily lymphocytes of unknown CD4/8 phenotypes) in cattle at DON levels as low as 0.21 µM have been observed [13,25]. While true absorption of DON in steers of this study is unknown, even a conservative estimate of blood volume (425 kg average body weight, 9 kg TMR intake/day [24]) would conclude that cytotoxic DON levels are reported to perturb PBMCs from this species [13] would be attainable only with a high absorption and distribution of the parent toxin. A previous work by Taranu (2010) reported that FUM was a very poor inhibitor of PBMC proliferation; however, the combined effects of DON and FUM on PBMCs in this species have not been explored. Taken together, these results suggest that the CD4:CD8 ratio could be a candidate biomarker of early DON exposure in beef cattle, which may be attenuated by co-occurring FUM toxicity. These results support previous literature citing the sensitivity of lymphocytes to this class of toxins [12,13,24]. Additional validation studies with a larger number of samples across more stratified doses are warranted to further explore this possibility. 

### 2.2. RNA-Seq 

Principal components analysis of the RNA-Seq data revealed that the third principal component exhibited a significant increase in the dietary treatment group compared with the control group (Bonferroni-adjusted *p* = 0.0207, Appendix A). This suggests that there were overall variation patterns in gene expressions caused by the dietary treatment. RNA-Seq analysis revealed 217 differentially expressed genes (DEG; FDR ≤ 0.05) between treatment- and control-fed steers, with mycotoxin exposure broadly leading to down-regulation of annotated features (Appendix A).

### 2.3. DIA Results

According to the DIA summary (Figure 4; Appendix A), the most impacted pathways were “Human Diseases,” followed by “Organismal Systems” and “Environmental Information Processing.” The high impact of those pathways was due to the significance of a few sub-categories of pathways, which all demonstrated inhibition in the treatment group. 

#### 2.3.1. Human Diseases

Among the Human Diseases sub-categories, “Endocrine and Metabolic Diseases” and pathways related to “Infectious Diseases” were the most impacted and were inhibited in the treatment group. It is unclear the reason for which the latter pathway was so significantly altered; however, we speculate that this may indicate a reduced immune activation/capacity. For “Endocrine and Metabolic Diseases,” the "AGE-RAGE Signaling Pathway in Diabetic Complications" was the most significantly inhibited and was among the most impacted pathways for all time points. This pathway is associated with response to oxidative stress and inflammation [26]; the results demonstrated here indicate an inhibition of cellular response to signaling cues in animals exposed to the DON/FUM diet which persisted throughout the clearance period.

#### 2.3.2. Organismal System

The high impact of the Organismal System was clearly due to a strong inhibition of the digestive system. The “Protein Digestion and Absorption” pathway was the most significant influence for this, as it demonstrated strong inhibition. The relevance of this pathway is unclear, since the samples analyzed were circulating blood cells and not tissues associated with the digestive system. The immune system was also highly impacted, showing strong inhibition for “Platelet Activation”. Platelets are important components for immune activation [27]; thus, inhibition may again indicate an overall trend of immune suppression.

#### 2.3.3. Environmental Information Processing

Environmental Information Processing was impacted due to strong inhibition from the “Signaling Molecules and Interaction” pathway. This was the consequence of inhibition of the “Extracellular Matrix (ECM)-Receptor Interaction”, while inhibition of “Cellular Growth and Death” was due to inhibition of the “Focal Adhesion” pathway. The P53 pathway was also inhibited, further indicating an alteration of cell cycle control. DON modulation of ECM receptor interactions has been previously noted in both directly exposed cell cultures [28] and, among other tissues, in the thymus of mice exposed to oral DON [29]. Among signaling pathways, the “PI3K-Akt Signaling Pathway” was strongly inhibited, while the “TGF-beta Signaling Pathway” was activated (see Appendix A). These alterations may indicate immune depression/suppression. The “PPAR Signaling Pathway,” an important pathway in immune differentiation and fate commitment [30], was also inhibited.

#### 2.3.4. Metabolism

No key metabolic pathways had clear outcomes related to treatment, with the exception of a small subset of pathways related to the metabolism of amino acids, including ‘Thiamine Metabolism’ (at both day 7 and 21) and ‘Valine, Leucine and Isoleucine Biosynthesis’ at day 21. While animals in this study had alterations in performance and digestive and circulating biochemical parameters (which indicated disruption of nutrient utilization [24]), it is difficult to determine if these disruptions were directly linked to the altered metabolism observed in circulating leukocytes. As these data point to a potential intervention in *Fusarium* toxin pathology, the link between DON/FUM exposure, amino acid metabolism and the systemic immune response warrants further investigation with additional exposure doses and treatment/clearance windows.

### 2.4. DAVID Results

Full DAVID analysis terms and clusters are provided in the Appendix A. 

#### 2.4.1. Oxygen Transport 

Numerous terms associated with heme and oxygen transport were identified from down-regulated genes at day 7, with this theme persisting throughout the entire treatment and clearance period. While these features were overshadowed by collagen and filament-associated terms in days 21 and 35, the sustained enrichment of these terms in down-regulated genes from treatment-fed steers indicates that iron utilization may be adversely affected by DON/FUM exposure. The disruptive and persistent effects of these toxins on the digestive system have been well reviewed [1,2], and this may be the mechanism by which they altered iron-dependent gene networks in the present study. To our knowledge, more direct changes in oxygen transport have not been previously characterized in this species; however, previous studies have shown alterations to cellular oxygen transport mechanisms by in vitro administration of fumonisins [1,16,17,18]. Given the rapid growth rate and large musculoskeletal systems of beef steers, the potential for these toxins to systemically alter oxygen transport may increase the risk of cardiovascular and hemolytic events in animals already experiencing the increased physiological burden of feedlot finishing growth state and is deserving of further investigation in animals under additional environmental stressors. 

#### 2.4.2. Extracellular Matrix and Signaling Pathways

Terms associated with the extracellular space and collagen organization were down-regulated in treatment-fed steers, with focal adhesion and the PI3K-Akt signaling pathway being enriched among down-regulated genes at all three timepoints measured. By day 21, terms associated with keratin, collagen and intermediate filament were amongst the most highly enriched clusters. Several of these clusters were also present in terms from down-regulated genes of treatment-fed steers during the clearance period (day 35). Associated physiological networks have been previously linked to fumonisin toxicity (via disruption of sphingolipid synthesis pathways) [1,16,17,18]. 

Of note is the consistent inhibition revealed by both tools (DIA and DAVID) for focal adhesion, ECM signaling, and the PI3K-Akt signaling pathway throughout the study period. Cell adhesion is crucial for optimal functioning of the immune system, especially for extravasation and migration [31]. The PI3K/Akt signaling pathway is pivotal, together with mTOR, for regulating the immune system, as these two proteins are involved in controlling the anti-inflammatory response of immune cells [32]. The role of focal adhesion is less clear; however, we hypothesize that it may be involved in T-cell migration [33].

#### 2.4.3. Inflammatory Response, Neutrophil Chemotaxis and Immunoglobulin

Immunotoxic effects of DON/FUM exposure were characterized by the upregulation of inflammatory response terms at days 7 and 21, along with neutrophil chemotaxis terms at day 7. Early toxin exposure was also accompanied by a numerical increase in neutrophil abundance (Figure 1); however, functional assays showed a slight (though insignificant) decrease in granulocyte phagocytosis and oxidative burst (Figure 2). These DEG clusters were not identified at day 35, and it is unclear if the gene expression changes observed here were incapable of resulting in altered performance, or if functional assays did not adequately capture these changes. Immunoglobulin-related pathways were enriched in animals exposed to DON/FUM at day 21. This change did not persist to day 35 and was not observed at day 7. The effects of DON on laboratory animal immunoglobulin levels has been previously reviewed [1,2], and has been shown to result in altered Ig production in a class-dependent manner. Furthermore, previous work has shown that disruptions to antibody production are an important nonlinear toxicological outcome in this species [8,12,13,14,15], a conclusion which these data further support. 

In general, the data indicate an overall effect on the transcriptome by the dietary mycotoxin treatment in beef cattle. The effect of the treatment on metabolism was negligible. It was interesting to note the consistent inhibition revealed by both tools (DIA and DAVID) for focal adhesion, ECM signaling, and the PI3K-Akt signaling pathway. Cell adhesion is crucial for optimal functioning of the immune system, especially for extravasation and migration [31]. The PI3K/Akt signaling pathway is pivotal, together with mTOR, for regulating the immune system; these two proteins are involved in controlling the anti-inflammatory response of immune cells [32]. The role of focal adhesion is less clear; however, it may be involved with T-cell migration [33].

## 3. Conclusions

This work aimed to quantify transcriptomic and immunological outcomes of multi-*Fusarium* toxin exposure in finishing beef cattle at levels similar to those encountered in production scenarios (1.7 mg DON and 3.5 mg FUM per kg ration). These results are applicable in both research (immunologists, nutritionists) and clinical (veterinarians, producers) settings, as the commercial value and previously established resistance of this species to DON and FUM indicates that the physiological state (finishing stage or end of production cycle) of these animals may play a role in these outcomes. Deviations in T-helper and cytotoxic cell profiles of treatment-fed steers, along with alterations to several immunomodulatory gene networks, indicate that the immune system of animals in this production class may be altered by *Fusarium* mycotoxin exposure when compared to control-fed animals of the same genetic background and production stage. Both DIA and DAVID analysis tools showed consistent inhibition of focal adhesion, ECM signaling and PI3K-Akt signaling pathways. Alterations in immune cell chemotaxis, immunoglobulin production and cell membrane structure gene networks were identified as important outcomes of exposure. Unresolved perturbations to cell membrane structure and oxygen and heme transport gene networks identified these as potentially chronic toxicological outcomes in this species. These results indicate that exposure to DON and FUM is detrimental to the welfare of finishing steers and may compromise their ability to withstand other stressors such as disease, heat stress or other toxins. In terms of energy utilization and growth, the clearance period used was appropriate to return treatment animals to performance indicators that were similar to control animals; however, a greater clearance period is needed to resolve gene regulation changes observed in beef cattle.

## 4. Materials and Methods

### 4.1. Study Design and Diet Formulation

All animal experiments were approved by the Oregon State University (OSU) Institutional Animal Care and Use Committee (Animal Care and Use Protocol #4986). In brief, 12 single-sourced Angus beef steers (mean body weight 419 ± 27 kg at first sample collection) were communally housed with individual access to assigned rations through a collar-transmitter gate system (American Calan, Northwood, NH, USA). Animals arrived at the research facility 43 days prior to dietary treatments (day −43; Figure 5), at which time each animal was evaluated by a veterinarian and treated metaphylactically with a single dose of Draxxin (Zoetis, Parsipanny, NJ, USA). Feed intake for the week prior to study initiation was used to block all animals to treatment. On day −4, animals were again evaluated by a veterinarian for overall health. Beginning on day 0, animals were fed according to their assigned dietary treatment (n = 6 animals per treatment).

Treatments differed in the ground corn constituent of the total mixed ration (TMR), which served as a source of natural toxin contamination, in addition to a fungal culture (produced by BIOMIN Research Center, Tulln, Austria). Average *Fusarium* toxin concentrations of final rations were 1.7 ± 0.2 mg for DON and 3.5 ± 0.3 mg for FUM per kg TMR for the treatment diet and 0.2 ± 0.1 mg for DON and 0.2 ± 0.2 mg for FUM per kg TMR for the control diet. FUM levels are reported as combined fumonisins B1, B2 and B3. Detailed mycotoxin analysis of TMR can be found in the Appendix A; this did not include examination for modified forms. These doses were selected to mirror those that beef cattle would ingest in a U.S. feedlot, where co-exposure of animals to multi-mycotoxins is common, but below guidance limits [3,5]. Throughout the study, TMRs from both groups were sampled each day, pooled by week, and sent to an accredited commercial laboratory for mycotoxin quantitation as described previously [24]; of the mycotoxins included in the panel, only nivalenol and ZEN were detected above the detection limit in the treatment samples (average 242 and 85 ug/kg). Aflatoxins, ochratoxin, acetyl-deoxynivalenol, fusarenon-X, HT-2 toxin, T-2 toxin, neosolaniol and diacetoxyscirpenol were below the limit of detection. Assigned diets were administered over the course of three weeks, after which time all animals were offered the control ration for a clearance period of two weeks (Figure 5). Additional diet formulation details may be found in the Appendix A. Each animal’s TMR was provided for ad libitum consumption at approximately the same time each morning. Drinking water was also supplied ad libitum. Feed consumption and other production data for these animals may be found in Duringer et al., (2020) [24].

### 4.2. Blood Collection

Blood was collected from each steer at approximately the same time of morning as outlined in Figure 5. Venous blood from a jugular vein was collected into BD Vacutainer^®^ (Becton, Dickinson and Company, Franklin Lakes, NJ, USA) collection tubes containing either EDTA (complete blood counts, CBC) or sodium heparin (flow cytometric analyses) as an anticoagulant and into Tempus^®^ (Thermo-Fisher Scientific, Waltham, MA, USA) blood collection tubes for RNA isolation. Upon blood collection, Vacutainer tubes were inverted gently several times and Tempus tubes were shaken vigorously for 60 seconds before being stored at room temperature for transport. Heparinized blood was completely processed (and cells fixed) within 4 h; EDTA anti-coagulated blood was immediately transported to an accredited veterinary diagnostic laboratory for CBC and Tempus tubes were stored at −80 °C for processing at the end of the trial. CBCs were performed on a Siemens ADVIA 120 Hematology System (Munich, Germany), with the white blood cell differential count performed manually using 100 cells. 

### 4.3. Reagents and Laboratory Conditions

For all assays described below, centrifugation was completed at room temperature, unless otherwise noted. Ultrapure water (18 mΩ) was acquired from an ELGA Ultra PureLab water purification system (Cary, NC, USA).

### 4.4. RNA Isolation, Sequencing, and Pathway Analysis

Total RNA was extracted using a Tempus™ Spin RNA Isolation Kit (Thermo-Fisher Scientific), per the manufacturer’s directions. Following isolation, all further RNA work was completed using the facilities at the OSU Center for Genome Research and Biocomputing (Corvallis, OR, USA). An Agilent 2100 Bioanalyzer (Agilent Technologies, Santa Clara, CA, USA) and an RNA Nano Chip (Agilent Technologies) were used to quantify RNA quality and concentration. All samples for cDNA library prep had satisfactory RNA Integrity Numbers of ≥ 8.0. 

Strand-specific cDNA libraries were prepared with poly-A selection for each sample, checked with an Agilent TapeStation 4200 (Agilent Technologies), and validated with single dilution qPCR. Each library was sequenced using an Illumina HiSeq 3000 (Illumina, Inc., San Diego, CA, USA) with a 100 bp read length (single end). Pseudoalignment of each cDNA library to the *Bos taurus* reference genome (UMD3.1.1) [34] was completed using kallisto v. 0.42.4 [35] and converted to gene-level estimates through the tximport R package [36]. 

### 4.5. Bioinformatics Analyses

Details of the Dynamic Impact Approach have been previously described [37]. Briefly, the DIA attempts to capture the biological impact of any condition as inferred through the combination of the proportion of DEG compared to all measured transcripts, the significance of the change of the DEG as –log_10_ of the P-value, and the log_2_ of the expression ratio to obtain an *impact*. This can then be interpreted as the overall impact of the condition studied on the genes associated with a particular term; in the present manuscript, only KEGG pathways were used. The DIA also delineates the direction of the impact (or *flux*), i.e., the overall direction of the impacted term (e.g., pathway) as being either activated or inhibited based on a combination of the expression ratio, significance of the change, and the effect of each protein coded by the affected genes on the specific term (e.g., pathway). For the current study, all annotated and measured genes were uploaded into the DIA using the following criteria: FDR cut-off = 0.05 and a minimum of four genes in each pathway. DIA functions through Excel (Microsoft, Redmond, WA USA) and can be requested from co-author Dr. Bionaz. An online version is available here https://dynamicimpactapproach.shinyapps.io/diarelease1/ (accessed on 15 January 2021).

The enrichment analysis of the DEG was performed using the Database for Annotation, Visualization and Integrated Discovery (DAVID) v.6.8 [38,39]. The whole annotated and measured transcriptome in the experiment was used as background and the default criteria were selected. The analysis was performed for each time point using all DEG (FDR ≤ 0.05), plus an analysis of only the up-regulated DEG and only the down-regulated DEG to be able to further interpret the results. From DAVID, the Chart and the Cluster list of the biological terms enriched with an EASE scope < 0.1 were downloaded.

### 4.6. Phagocytosis and Oxidative Burst

Modifications to a commercially available kit (pHrodo™ Green *E. coli*, Thermo-Fisher Scientific) were employed to simultaneously evaluate phagocytosis capacity and to differentiate populations of phagocytosing cells (see Appendix A for detailed protocols). Oxidative burst activity of immune cells was assessed following activation by Phorbol 12-myristate 13-acetate, again with modifications to a commercially available kit (Neutrophil/Monocyte Respiratory Burst Assay, Cayman Chemical, Ann Arbor, MC, USA). The red blood cell lysis and antibody incubation steps were adopted from the phagocytosis assay described above. 

The sample data was acquired on a CytoFLEX S (Beckman Coulter, Brea, California, USA) flow cytometer using CytExpert software v.2.1 (Beckman Coulter), while data transformation, quality control and gating were performed using FlowJo v.10.4.2 (FlowJo LLC, Ashland, Oregon, USA). Daily quality control beads (Beckman Coulter), the FlowClean R package [40] and manual gating of singlet events (SSC-A/SSC-H) were implemented for flow cytometry quality control. For comparisons of cell populations participating in phagocytosis and oxidative burst, all samples were compared against control samples using the Overton population comparison feature of FlowJo [41]. CD4^+^ and CD8^+^ cells were manually gated from lymphocytes identified as nucleated, singlet, non-granulocyte, low FSC/SSC events (Appendix A), with gate positions set against control samples. 

### 4.7. Data Handling and Statistical Analysis

Statistical analysis of the immunotoxicity data was performed using R v.3.6 [42] and a linear regression model for each of the non-baseline time points. Outliers were removed (1.5 × IQR), and the final model included the baseline values (for repeated measures) and diet as main effects. RNA-Seq gene-level abundance data were first analyzed using principal components analysis to assess the statistical significance of systematic patterns of gene expression variations in association with the dietary treatment. Variance stabilizing transformation was performed prior to PCA to normalize the gene expression counts. A linear mixed effects model was then fitted on each of the top three principal components (PCs), with dietary treatment and time included as explanatory variables. To detect individual DEG, the RNA-Seq data were analyzed using negative binomial models for each of the non-baseline time points, correcting by each animal’s day 0 values. Differential expression analysis was performed using quasi-likelihood tests with the limma and edgeR R packages [43,44,45]. To adjust for multiple testing, the Benjamini-Hochberg (BH) procedure was performed for treatment period samples (weeks 1–3) separately from clearance period samples (weeks 4–5). As the aim of this research was to differentiate a large number of outcomes, which may be even subtly altered by the toxin doses used, differences were considered significant at an adjusted *p*-value of ≤ 0.10, which controls the false discovery rate (FDR) at 0.10.

## Figures and Tables

**Figure 1 toxins-13-00714-f001:**
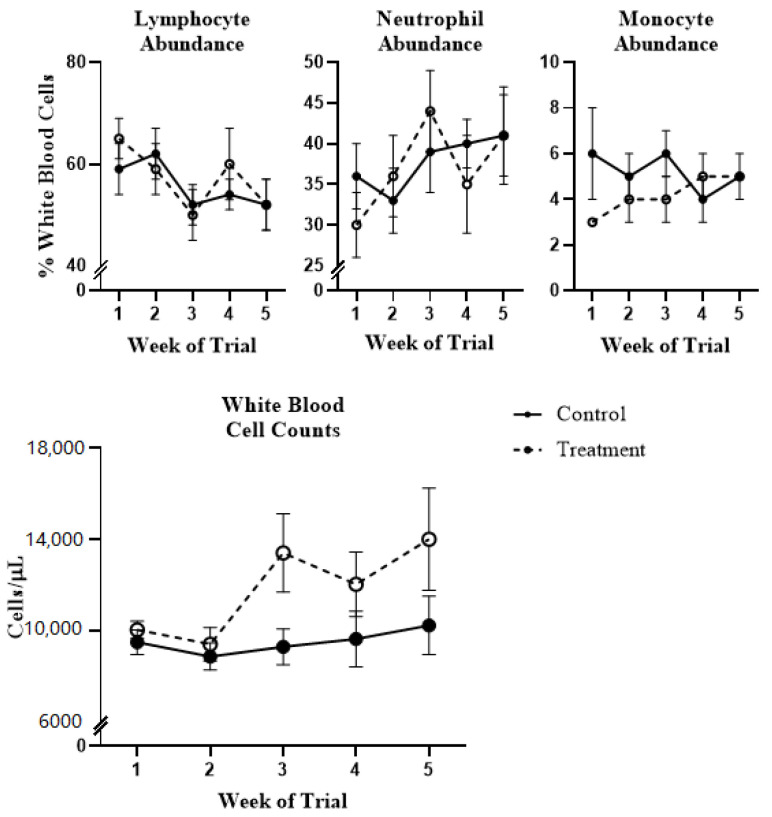
White blood cell differential (lymphocyte, neutrophil and monocyte abundance as a percentage of white blood cells, total white blood cells/µL blood) of beef cattle exposed to *Fusarium* mycotoxins for 21 days, followed by a 14-day clearance period. Values are observed means ± standard errors with significance from model described in the materials and methods. Reference values provided by diagnostic laboratory for these tests: white blood cell counts 4000–12,000/µL, with lymphocytes 45–75%, monocytes 2–7% and neutrophils 15–45% of white blood cells.

**Figure 2 toxins-13-00714-f002:**
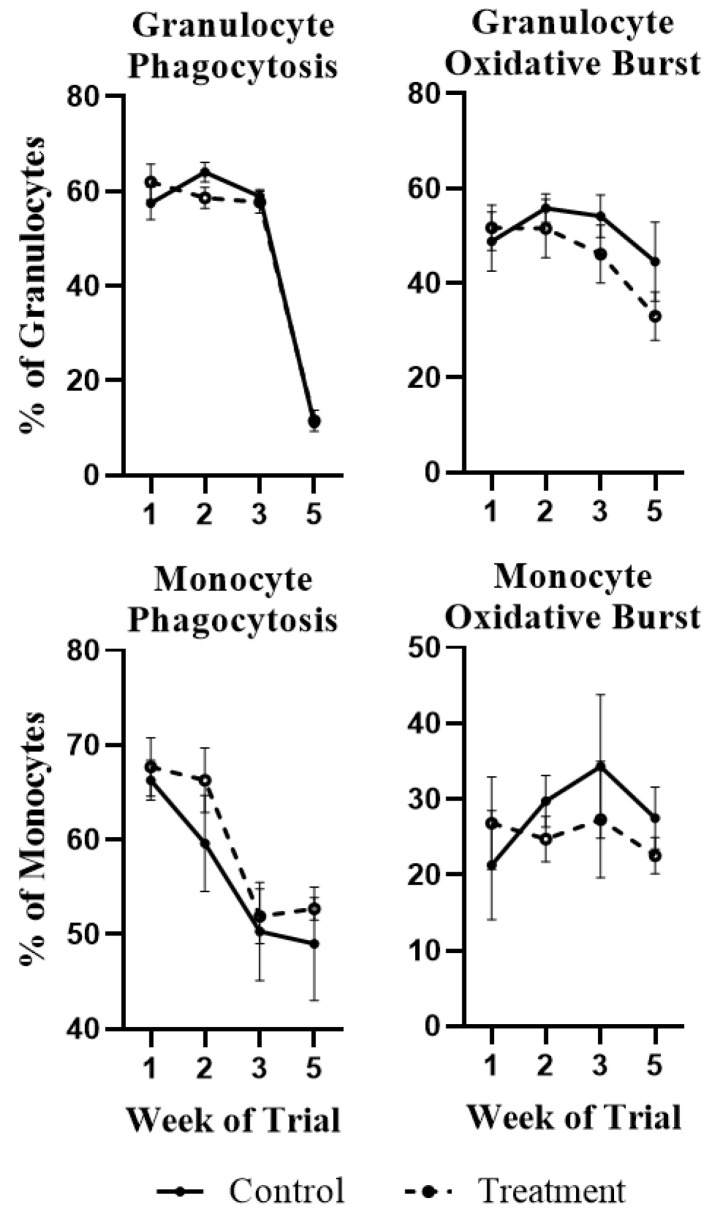
Phagocytosis and oxidative burst in granulocytes and monocytes of beef cattle exposed to *Fusarium* mycotoxins for 21 days, followed by a 14-day clearance period. Values are observed as means ± standard errors with significance from model described in the materials and methods.

**Figure 3 toxins-13-00714-f003:**
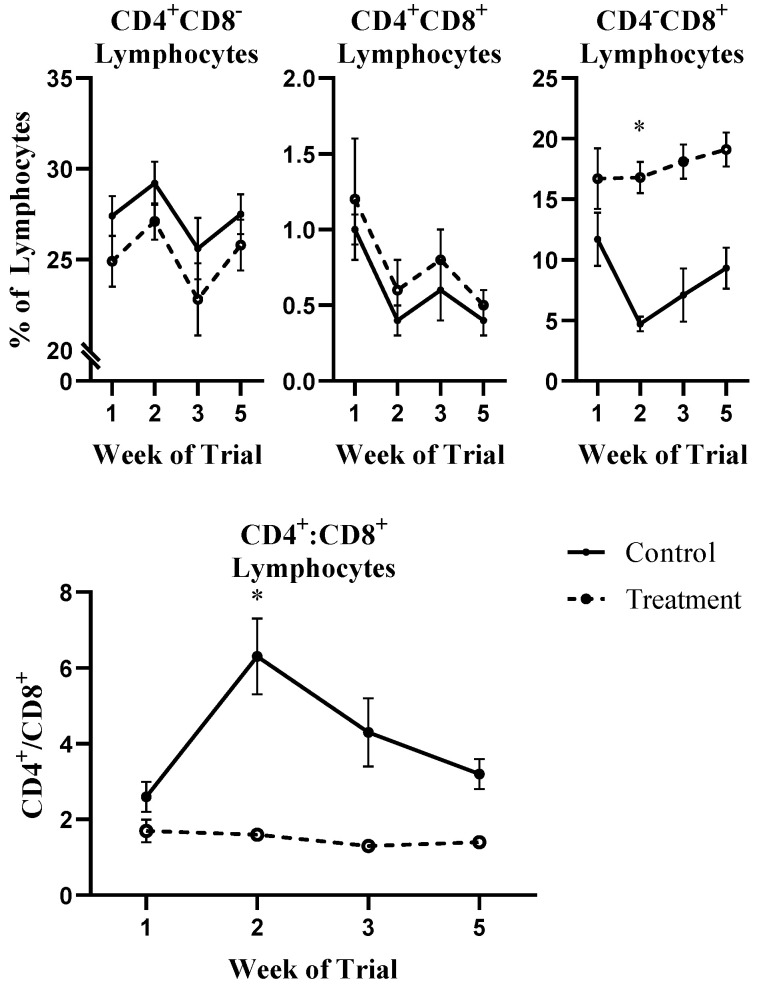
Lymphocyte CD4 and CD8 expression in beef cattle exposed to *Fusarium* mycotoxins for 21 days, followed by a 14-day clearance period on. Values are observed means ± standard errors with significance from model described in the materials and methods. * Adjusted p-value ≤ 0.10.

**Figure 4 toxins-13-00714-f004:**
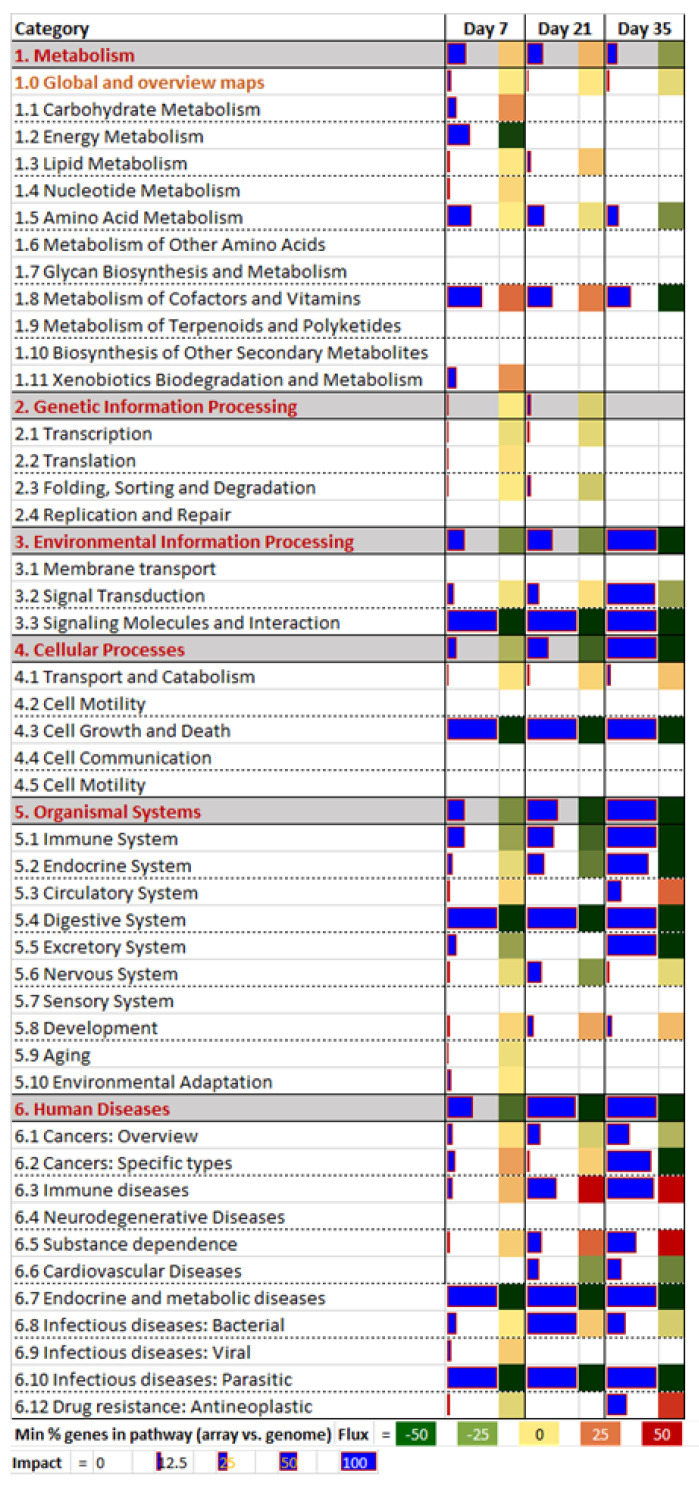
DIA results of the summary of Kyoto Encyclopedia of Genes and Genomes (KEGG) pathways affected by differentially expressed genes (DEG). Impact denotes the overall impact of the DEG on the pathways (size of blue bar) and the Flux denotes the overall direction of the impact (i.e., red = activated; dark green = inhibited; orange-light green = moderate change).

**Figure 5 toxins-13-00714-f005:**
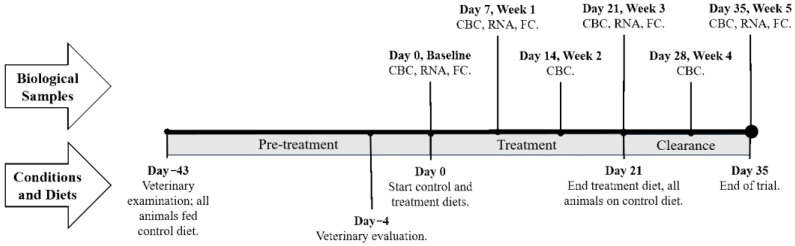
Timeline of experiment used to evaluate the effects of deoxynivalenol and fumonisins fed in combination on beef cattle. CBC = Blood collection for complete blood count; RNA = Tempus Tube blood collection (for RNA-Seq analysis); FC = Blood collection for flow cytometric analysis.

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
