# Peer review of "Effects of Deoxynivalenol and Fumonisins Fed in Combination to Beef Cattle: Immunotoxicity and Gene Expression"

_toxins, 2021, doi:10.3390/toxins13100714_

Round 1
Reviewer 1 Report
The paper Effects of Deoxynivalenol and Fumonisins Fed in Combination on Beef Cattle: Immunotoxicity and Gene Expression is a well designed and written paper, and there are few minor corrections needed prior the publication. All data are nicely explained and emphasised where results are clear, and where authors’ hypothesised or even suggested that analysis did not adequately capture changes. It is nice to see this in the manuscript so readers can understand the results with higher confidence.
In the abstract or within the paper please define which fumonisins were included in the study (FB1,FB2, FB3, FB4, FB6, FA1, FA2, FA3,…), and if there was a fumonisins mixture what was the ratio and why did you choose this ratio?
Line 34 – please add also the data on the occurrence of masked/modified mycotoxins in feed and food/by-products https://doi.org/10.3920/WMJ2012.1410; http://dx.doi.org/10.3390/toxins11010030; and add explanation/comment on the occurrence of masked/modified mycotoxins in your study (did you check for their occurrence, or ensure that they would not form) (https://dx.doi.org/10.1002%2Fmnfr.201100764; https://dx.doi.org/10.2478/aiht-2018-69-3108).
Please explain in the text chosen concentrations of the mycotoxins in this study – based on the natural mycotoxin occurrence, or calculated theoretical toxicological effects or something third. Please try to back up the explanation by other publications.
In the conclusion, please add one sentence on the applicability of the results of the study
Please explain abb. on the first use (KEGG).
Line 288 – please add (in supplementary file) detailed mycotoxins concentrations in the control and intervention feed with the respective LOD’s of the analysed mycotoxins
Line 293 – in the Duringer et al., (2020) there is limited mycotoxin data available in the cited reference, please provide full mycotoxin analysis details here, and the results for the control and intervention feed in the supplementary materials. (together with line 288)
Author Response
Reviewer 1
The paper Effects of Deoxynivalenol and Fumonisins Fed in Combination on Beef Cattle: Immunotoxicity and Gene Expression is a well designed and written paper, and there are few minor corrections needed prior the publication. All data are nicely explained and emphasised where results are clear, and where authors’ hypothesised or even suggested that analysis did not adequately capture changes. It is nice to see this in the manuscript so readers can understand the results with higher confidence.
- The authors thank you for this response.
In the abstract or within the paper please define which fumonisins were included in the study (FB1,FB2, FB3, FB4, FB6, FA1, FA2, FA3,…), and if there was a fumonisins mixture what was the ratio and why did you choose this ratio?
- This information has been added to the abstract (FB1, FB2, FB3).
- The results of the mycotoxin profile analyzed have been added to the attached supplementary materials and mentioned in the materials and methods.
- The fumonisin ratio used was naturally occurring in the contaminated corn and supplementary culture material and was not further manipulated or altered.
- We have added an additional line to our introduction to emphasize the decisions behind the mycotoxin levels used in this study.
Line 34 – please add also the data on the occurrence of masked/modified mycotoxins in feed and food/by-products https://doi.org/10.3920/WMJ2012.1410; http://dx.doi.org/10.3390/toxins11010030; and add explanation/comment on the occurrence of masked/modified mycotoxins in your study (did you check for their occurrence, or ensure that they would not form) (https://dx.doi.org/10.1002%2Fmnfr.201100764; https://dx.doi.org/10.2478/aiht-2018-69-3108).
- We did not examine for the presence of masked mycotoxins, as that is not current practice for mycotoxin analysis reports obtained from livestock operations when evaluating feed sources. This was clarified in the materials and methods.
Please explain in the text chosen concentrations of the mycotoxins in this study – based on the natural mycotoxin occurrence, or calculated theoretical toxicological effects or something third. Please try to back up the explanation by other publications.
- Explanation for dose chosen was added with references in introduction, last paragraph, as far as targeting a dose that was relevant to typical beef cattle feed regimen but one that was not above a toxic threshold.
In the conclusion, please add one sentence on the applicability of the results of the study
- Please see revisions to conclusion.
Please explain abb. on the first use (KEGG).
- Completed (in caption of Figure 4).
Line 288 – please add (in supplementary file) detailed mycotoxins concentrations in the control and intervention feed with the respective LOD’s of the analysed mycotoxins
- Added in supplementary materials, including LODs.
Line 293 – in the Duringer et al., (2020) there is limited mycotoxin data available in the cited reference, please provide full mycotoxin analysis details here, and the results for the control and intervention feed in the supplementary materials. (together with line 288)
- See previous comment.

Reviewer 2 Report
Dear authors,
The article presented for review raises an interesting and important issue of the influence of mycotoxins on the health of beef cattle.
First of all, I must commend a very well-planned experiment on animals
However, I would like to point out a few problems:
The results of blood morphology and flow cytometry indicate no effect of DON and FUM on the examined tissue (except for CD8 lymphocytes)
Why was the significance level of the statistical tests set at p = 0.1 and not p = 0.05?
This article talks about fumonisins, but which FB1, FB2?
There is no information about the basic composition of the feeds used – protein, fat, etc.… .
There is no information about the TMR component composition
Lack of production results: weight gain, feed consumption, etc ...
Best regards
Author Response
Reviewer 2
The article presented for review raises an interesting and important issue of the influence of mycotoxins on the health of beef cattle. First of all, I must commend a very well-planned experiment on animals. However, I would like to point out a few problems:
The results of blood morphology and flow cytometry indicate no effect of DON and FUM on the examined tissue (except for CD8 lymphocytes)
- A sentence has been added to the results section (section 2.1) to specifically highlight the main finding of the blood parameter and support this finding with literature references.
Why was the significance level of the statistical tests set at p = 0.1 and not p = 0.05?
- This has been added to the materials and methods (section 4.7).
This article talks about fumonisins, but which FB1, FB2?
- This has been added to the abstract, the materials and methods (section 4.1) and the supplementary materials.
There is no information about the basic composition of the feeds used – protein, fat, etc.…
- Please see additional information now provided in supplementary materials.
There is no information about the TMR component composition
- Please see additional information now provided in supplementary materials.
Lack of production results: weight gain, feed consumption, etc …
- These values have been previously published. An additional reference to this publication has been added to the materials and methods (section 4.1).

Reviewer 3 Report
This manuscript evaluates the effects of two mycotoxins on cattle health and immune response. In the context of current co-contamination by several mycotoxins, it’s important to be able to evaluate the effects induced and to try to find accurate biomarkers of the animal health status. I have several comments.
- L36-38: Can you cite some examples of toxicity that can be observed on cattle exposed to mycotoxins. We miss a little of information to really understand the importance and the problem of mycotoxins contamination in cattle.
- Can you increase the amount of details in captions, not enough details mentioned.
For example, in figure 1, if I’m correct, you start to describe the figure at the bottom instead of the three on the top, and it’s not clearly mentioned on the figures that it’s Total white blood cells. Maybe some letters can be added.
- Are the percentages of the three main leukocytes types in a normal range for cattle?
For the results why don’t you display the week 4 - day 28, if I’m following your timeline scheme, I can see that you collected the blood as for week 2- day 14, which one is described.
- For the figure 4, explain more in details your figure in the caption, what mean all the different squares and color. Remind the meaning of impact and what means “array vs. genome”. At the first sight, the figure is not so easy to read.
Supplementary materials part:
- The text in the supplementary materials part is not merged.
- In tables S1 and S2, can you indicate the secondary antibody used.
- I miss some details about the cytometry part. Can you detail the gating strategy used un Suppl. Fig.2
- The numbers in the sentence “see attached excel file (Supp_Tables_1)” doesn’t fit with the name of the table in which it is mentioned, Table S3,S4, S6_S14, S15-S23 and S24-S26, is it normal?.
- The figure S1 is not understandable if it’s printed in black and white. Also, it’s not mention in the caption that it’s an ACP, neither the percentages of the axis. Usually, Axis 1 and 2 are displayed not 2 and 3.
- For the figure S2, I would like to see visually all the gating strategy. Did you only do FSC/SSC then CD4/CD8?
Author Response
Reviewer 3
This manuscript evaluates the effects of two mycotoxins on cattle health and immune response. In the context of current co-contamination by several mycotoxins, it’s important to be able to evaluate the effects induced and to try to find accurate biomarkers of the animal health status. I have several comments.
L36-38: Can you cite some examples of toxicity that can be observed on cattle exposed to mycotoxins. We miss a little of information to really understand the importance and the problem of mycotoxins contamination in cattle.
- Information specific to DON and FUM exposure were added for cattle toxicity in addition to a more general statement on the need for such research.
Can you increase the amount of details in captions, not enough details mentioned. For example, in figure 1, if I’m correct, you start to describe the figure at the bottom instead of the three on the top, and it’s not clearly mentioned on the figures that it’s Total white blood cells. Maybe some letters can be added.
- We have added additional details to the figure captions.
Are the percentages of the three main leukocytes types in a normal range for cattle?
- Please see changes made to page 2 (results and discussion) and 3 (figure caption) of manuscript. Note that these results (CBCs) were obtained from a veterinary diagnostic laboratory, thus the reference values used by that facility are included here.
For the results why don’t you display the week 4 - day 28, if I’m following your timeline scheme, I can see that you collected the blood as for week 2- day 14, which one is described.
- Figure 1 has been updated to include the requested data, which was excluded in error.
- During the course of this correction, we realized that the entire plotted data set for the data represented in this figure (WBC count, lymphocyte %s, monocyte %s, granulocyte %s) had been shifted over by one week, while one week (week four) was excluded. This required only a minor change to the manuscript text, which has been updated. As our core conclusions were drawn from raw data (and this error affected only the plotted values), this change did not result in changes to our interpretation of this data. We apologize for any confusion surrounding this issue, and thank the reviewer for catching the missing data points which ultimately led to our discovery of a larger issue for this figure.
For the figure 4, explain more in details your figure in the caption, what mean all the different squares and color. Remind the meaning of impact and what means “array vs. genome”. At the first sight, the figure is not so easy to read.
- Additional explanation added to the caption for figure 4.
Supplementary materials part:
The text in the supplementary materials part is not merged.
- We are unclear what is meant by this comment but are hoping that other changes to the supplementary materials section of this paper have altered this section to meet the reviewers needs. We are glad to alter again if needed.
In tables S1 and S2, can you indicate the secondary antibody used.
- Please see adjusted tables.
I miss some details about the cytometry part. Can you detail the gating strategy used un Suppl. Fig.2
- This figure has been altered to show upstream gating used for the CD4/CD8 lymphocyte differential. Unfortunately, we did not note the exact sample used for the original graphic, thus we have chosen a separate sample to generate the complete set of graphics now shown in this figure.
The numbers in the sentence “see attached excel file (Supp_Tables_1)” doesn’t fit with the name of the table in which it is mentioned, Table S3,S4, S6_S14, S15-S23 and S24-S26, is it normal?.
- The naming schema of the supplementary materials has been altered to reflect this change.
The figure S1 is not understandable if it’s printed in black and white. Also, it’s not mention in the caption that it’s an ACP, neither the percentages of the axis.
Usually, Axis 1 and 2 are displayed not 2 and 3.
- The corresponding author attempted to contact the author who originated this figure but was unable to reach them. The corresponding author will continue to reach out and, if the editor would like this figure changed, will make every effort to produce a figure in only black and white and to update axis as appropriate. With regards to the color, we have another figure in this manuscript that will also be difficult to understand if it is printed in black and white (Figure 4); thus we would like to proceed with the final review of our manuscript at this time with this figure in color.
- Given the overall goal of our study design (to evaluate the effects of DON/FUM inclusion on measured outcomes), we felt these components most appropriate to include in the final presentation of this data, as PC3 showed differentiation in the data with a significant dietary effect (discussed briefly in section 2.2 of the manuscript). Clustering of diet against PC1/2 was inconclusive, and we did not feel that PC1/2 comparisons added value to the manuscript as a whole.
- While we feel the significant dietary effect on PC3 is an important finding of the work presented here, this has not led to conclusions which we have not supported via other means within the manuscript. Thus, we are willing to remove PCA from these findings in favor of moving forward with the remainder of the manuscript, if the reviewer prefers.
For the figure S2, I would like to see visually all the gating strategy. Did you only do FSC/SSC then CD4/CD8?
- Please see altered figure.
